# Psychological Skills Training for Athletes in Sports: Web of Science Bibliometric Analysis

**DOI:** 10.3390/healthcare11020259

**Published:** 2023-01-13

**Authors:** Inchon Park, Jonghwan Jeon

**Affiliations:** Department of Sports Convergence, Kyungil University, Gyeongsan 38428, Republic of Korea

**Keywords:** psychological skills, psychological sport, psychological training, mental training, sport, performance

## Abstract

Psychological Skill Training (PST) for optimal performance in sports has received extensive attention from athletes, coaches, and experts, and numerous studies have been conducted, but despite this interest, it has not been the subject of any bibliographic analysis. The analysis covered 405 articles and reviews that were located in the Thomson Reuters Web of Science (Core CollectionTM) between 1992 and 2021. These papers were published by 1048 authors from 543 different universities across 54 countries in 107 different journals. The fundamental bibliometric calculations and co-word networks were completed. As a result, the following thematic elements were grouped into four clusters: (1) PST for stress, mental toughness, and coping, (2) PST for anxiety, motivation, self-confidence, and self-efficacy, (3) PST for flow and mindfulness, and (4) PST for emotions. From the results of this study, it was confirmed that the topic of PST is converging toward the best performance, with various approaches, such as stress management, anxiety control, and coping as techniques for PST. Furthermore, mindfulness and positive psychology studies emphasize athletes’ quality of life, so coaches and experts should pay more attention to improving athletes’ quality of life in future research.

## 1. Introduction

In today’s Olympics and international level competitions, the physical, technical, and strategic gaps between top athletes are progressively shrinking. It was amazing to witness that certain athletes performed better than athletes who were more physically talented. Conversely, it was disappointing that some athletes would perform flawlessly during the semi-final and then perform tragically in the final. The massive pressure from this increased performance density has been directly linked to the extensive attention to psychological skills training (PST) in recent. In the early days, coaches and athletes recognized the importance of mental states for optimal performance, but the field of sports psychological training was not flourished because of the misunderstanding that psychological skills are innate properties and lack of knowledge to train these abilities [1,2].

Initially, sport psychology or mental training with athletes emerged in the United States during the late 1970s. Although one of the first works in sport psychology was published in the 1920s by Coleman Griffith [3], it went through a long hibernation until the 1960s and early 1970s, when systematic studies on sports psychology began [1,4]. In the late 1980s, Vealey suggested that future directions in PST include expanding the target population, refining specific implementation procedures, and differentiating between psychological skills and methods [1]. During the first 7 years of the 1980s, the premier scientific publications, such as the Journal of Sport Psychology, mainly focused on theoretically based research that emphasized various psychological approaches and research methodologies [4,5]. While several experts acknowledged that applied sport psychology interventions could be effective through their reviews [6,7], they urged greater attention must be given to program evaluation and professional accountability. With these accountability concerns, research evolved and concentrated mainly on evaluating the effectiveness of psychological services to athletes and coaches [8,9]. The Consultant Evaluation Form (CEF) by Partington and Orlick [10] was the first valid and reliable instrument to evaluate the qualification of psychological consultants and their services. With the efforts to identify the effective sport psychology consultant characteristics [8,11] and the publication of psychological consultant guidelines for athletes [12], recent studies have focused primarily on interventional research [13,14].

Although the field of PST covering all sports events has produced a copious number of publications, the attempts to gather bibliometric data in a systematic manner to identify research trends and highlight publications that have led to the progress of the field have not been done yet. During the 1980s and early 1990s, there have been few bibliometric studies in sports and exercise science, such as the International Society of Sport Psychology [15], the Journal of Sport Psychology [4], and Sport & Exercise Psychology [16]. However, it was analyzed only within a specific journal and did not include all journals in the related field.

The bibliometric analysis provides information including highly cited authors, publications, the most productive institutions, and countries through the analysis of citation indices. Bibliographic studies emphasize trend-driven authors in the research field and provide information on the current topics; they have been widely applied to a variety of other research areas and have provided valuable insights [17,18]. Thus, it allows researchers to determine the scope of research topics and assists them in planning their research direction and predicting research trends. Several bibliometric studies have investigated the publication trend of sport and exercise science concerning the research output of particular countries or areas [19,20], sub-specialized fields [21,22], or the production of a selection of sport science journals [23]. To the best of our knowledge, there are no bibliometric analyses that have investigated research regarding PST. This study used bibliometric tools to analyze PST articles retrieved on the Web of Science (Thomson Reuters Company, Toronto, ON, Canada) database and provides a better understanding of how PST research was initiated, organized, progressed, and interrelated throughout the world. In addition, it aims not only to provide insights into future research for experts in the academic field but also to provide practical help in the development of PST programs for coaches and athletes in the sports field.

## 2. Materials and Methods

The Thomson Reuters Web of Science (WoS) database’s bibliometric data for this study was obtained on 10 November 2021. Prior to data collection, five researchers in the field of sport and exercise psychology were asked to indicate search terms they would use to retrieve publications on PST and elite athletes. For the search, the terms “mental skill*,” “Psychological skill*,” “mental skill training*,” and “psychological skill training*” were used in the topic search field without limitation on the publication year of the documents. In order to retrieve publications that specifically focused on athletes or elite athletes, we refined our search with the following search terms “sport*,” “athlete*,” “elite*,” and “elite athlete*.” The topic field searches the title, abstracts, author’s keywords, and KeyWords Plus^®^ for the terms provided (keywords automatically assigned by WoS). Boolean operators (AND-OR) were also utilized to enhance the search for associated publications. Only journal articles and reviews were used for the analysis because they attributed the majority of document types and consisted of complete research ideas and results; therefore, 19 meeting abstracts, 14 early access articles, 4 book reviews, 4 proceedings papers, 3 editorial materials, and 2 book chapters were excluded. In the end, a total of 405 related articles and reviews were analyzed in the current research.

For analysis, the data were downloaded from WoS in “Full record and cited references” and “plain text” formats. First, to identify the authors, countries, journals, and institutions with the largest number of articles and citations, HISTCITE 12.3.17 (Thomson Reuters, Philadelphia, PA, USA) software tool was used to analyze the data. Qualitative indexes [global citation score (GCS) and local citation score (LCS)] were considered in this study [24]. The GCS and LCS metrics quantify the number of citations over the whole Web of Science Core Collection and the identical collection, respectively [25]. The co-word networks between the keywords were then examined using VOSviewer’s methodology [26]. The linkages and networks between the keywords were found and analyzed using this program.

To create a visualized map for the bibliographic analysis, we imported the downloaded data into VOSviewer. It enables us to select and adjust settings in accordance with various analytic objectives and data sources, as data cleaning is frequently required when producing maps based on web data. As a result, the following criteria are established for this study. (a) It is possible to combine or ignore certain terms when creating mappings based on text data by using the thesaurus files supplied by VOSviewer. For instance, for a more precise clustering analysis, the terms “coping,” “coping skill,” and “coping skills” were combined using the thesaurus file, while terms irrelevant to this study that were not explicitly filtered out, such as “coping skill therapist,” were not omitted. (b) The strength of the connection between normative items was determined using the association strength method [26], which was deemed to be the most consistent with the normalized technique. (c) Following testing, the layout with the parameter of attraction set to 2 and the parameter of repulsion set to 0 (producing a map of the co-word network) produced the best visual results. Additionally, the default settings for the other options.

Figure 1 denotes the step-by-step processes of this study: (Step 1) The topics and keywords related to PST/mental skill training and athlete/elite were identified, and then the search was defined. (Step 2) We refined the searched items and eliminated the irrelevant items. Finally, 405 articles were included. (Step 3) We sorted the papers into categories according to the year, author, number of citations, journal, country, and institution. (Step 4) The co-word map was generated. The information in the networks was examined in Step 5 to determine the outcomes.

## 3. Results

After the data were refined for this analysis, a total of 405 papers from 107 different journals between 1992 and 2021 were examined. As seen in Figure 2, publications increased steadily between 2007 and the present, with 83.45% (*n* = 338) of the total papers appearing in the previous fifteen years (2007–2021), while only 67 articles (16.54%) had been published from 1992 to 2006. In some research fields, the era of an extremely low number of articles (Since 1992, five articles on average have been published per year) might cause it to be regarded as a “niche” academic field [27]. The number of publications peaked in 2019, while the number of published articles in 2020 decreased by 35% compared to the previous year (Figure 2).

Based on the results, PST in sports has recently drawn the interest of academics and professionals; however, it is impossible to tell for sure whether this trend will continue in the future. According to Price’s law [28], the research process, however, goes through four stages: (i) pioneers begin publishing on a specific research field, (ii) due to the interest of many academics in the study topic, there is exponential development, (iii) a concentration of information and research on the subject, and (iv) a decrease in publications. It may be argued that PST is now a topic of interest for academics and professionals due to the aforementioned procedure. The impact of the COVID-19 pandemic, which forced coaches and athletes to train remotely and restrict the chance to meet a sport psychologist and practitioner, may have contributed to the decrease in publications in recent years, despite the fact that the number of related publications has decreased compared to 2019. The data acquired demonstrate that the number of publications in 2021 increased compared to the previous year, and the topics of PST-related research are becoming more diverse [29,30].

### 3.1. Authors and Number of Citations

There were a total of 1048 authors across the 405 publications that this study examined, hailing from 54 countries and 543 different institutions. Table 1 lists the authors who have published the most on PST in the sports area. The maximum number of publications to identify an author’s output in the topic area was the criteria used to rank the entries in Table 1 and Table 2, and (ii) the highest number of citations, which are widely used to evaluate the significance [31] of articles and researchers.

As a result, Andrew M. Lane, who has thirteen papers and a total of 411 citations in WoS, was named the most productive author (GCS). This author has published articles mainly related to emotion regulation and emotional intelligence for athletes in sports: develop a questionnaire and intervention regarding emotion regulation for the performance of endurance athletes. Second, Dave Collins has published ten articles with a total of 295 citations in WoS. The author developed the Psychological Characteristics of Developing Excellence Questionnaire. Finally, Tracey J. Devonport has published nine articles with a total of 294 citations and also explored the emotional regulation of athletes.

Even though some authors are more productive than others, this field of study can be considered fragmented because there is no clear “reference author”. Thus, Table 2 presents the most prominent authors, taking into account the number of citations, with publications related to PST in sports partially consistent with the highest number of authors. Despite the relatively small number of publications, Patrick R. Thomas recorded a high citation score and developed the Test of Performance Strategies to evaluate the strategies and psychological skills used by athletes [32]. Daniel Gould’s Studies examining the influences of mental skills and strategies on Olympic performance are also listed because they have a large number of citations.

**Table 2 healthcare-11-00259-t002:** Authors with the highest number of citations (>350).

Author	Affiliation	No.	LCS	GCS
Hanton, S	Cardiff Metropolitan University (UK)	8	75	439
Thomas, PR	Griffith University (Australia)	4	102	422
Lane, AM	Wolverhampton University (UK)	13	23	411
Gould, D	Michigan State University (USA)	4	45	400
Hardy, L	Bangor University (UK)	6	103	367

No.: number of articles, LCS: local citation score, GCL: global citation score.

### 3.2. Institutions

PST has been covered in articles by 543 different institutions. The institution is the one to which the researcher is affiliated at the time the article is published. More than 88% (481 out of 543) of the institutions have published only one or two articles. Moreover, institutions that published more than three articles but fewer than seven accounted for less than 2% (53 out of 543) of the total. The nine institutions that are presented in Table 3 have published more than seven articles.

Secondly, the University of Wales Institute Cardiff stands out in first place among institutions with the most citations across the whole WoS (GCS = 472), followed by Wolverhampton University (GCS = 432). Interestingly, although it did not list in Table 4 because of a small number of articles, Griffith University (GCS = 457) and the University of Western Ontario (GCS = 323) published four papers that recorded a high citation score and took second and fourth place, respectively. As confirmed by the number of citations and authors, research institutes in the UK are actively conducting research related to PST.

### 3.3. Journals

At least one paper on this subject has been published in 107 different journals. Journals accounting for more than 56% (60 out of 107) of the total have only published one paper, whereas journals making up more than 20% (22 out of 107) of the total have published two or three. Table 4. lists journals that have published more than 10 articles. Sport Psychologist, the most traditional journal in the PST field, published 58 articles and recorded the highest number of publications and GCS scores, but the recently reported impact factor was low at 1.45 (Q4). On the other hand, despite the small number of articles and low GCS scores, the Frontiers in Psychology and Journal of Sports Science and Medicine were reported as high-impact factors, which reflects the latest publishing trend that shows the option of open-access and a broad aim and scope of the journal.

### 3.4. Co-Word Analysis

In the field of research, keywords play a crucial role since they can be used to track the development of a specific area of knowledge. [33]. In the current study, 1076 keywords (both those chosen by the authors and by ISI WoS) were found, although only 428 (39.77%) of them co-occurred or showed up more than once. Of these, 60.22 % (or 648) were repeated just once. The term “co-word analysis” denotes a close association between the ideas and is defined as “a content analysis approach that employs the words in documents to identify relationships and develop a conceptual structure of the domain” [34]. Figure 2 shows the main co-occurrence connections found in the articles under investigation.

The most cited keywords are listed in Table 5; the most cited keyword was performance (GCS = 3333), followed by psychological (GCS = 2298), sport (GCS = 2157), and skill (GCS = 2121; Table 5). The keywords in Table 5 were chosen based on the following criteria: global citations in WoS (GCS) equal to or higher than 1000 citations for the most cited keywords and a frequency of presence in the search collection equal to or greater than 50 times for the most common keywords. However, compared to the most frequently used keywords, the keywords that receive more citations are slightly different.

Currently, powerful analytic tools such as VOSviewer allow for the systematic identification, analysis, and representation of keywords. Based on bibliographic information, a map was made to display a co-word network. The “association strength” method, the “Visualization of Similarities” (VOS) approach, was used to graphically organize each term on the map after the “association strength” method was used to standardize the association values of the keywords [35,36]. Finally, the VOSviewer method offers the option to incorporate several resolution settings in order to detect the various clusters. The keywords picked by the authors and those picked by the ISI WoS, as well as every other term in the whole list of keywords, were taken into consideration while conducting the thematic analysis. The cut-off point was established in 10 or more occurrences of these keywords. In this study, we ultimately settled on 45 keywords, and we assessed the overall strength of the co-occurrence links with other keywords. According to the analysis, the leading four different clusters of keywords were found. Figure 3 displays a graphical depiction of the co-occurrence of keywords or co-words.

This provides a generalized description of the knowledge or concepts found in previous works of literature [37]. Various sizes and colors of circles serve as representations for the analysis of the terms. A given keyword’s frequency is determined by the size of the circles; the larger the circle, the more times the term appears in the titles and abstracts of the publications under examination [38]. The clusters identified by the study correspond to the circles that can be distinguished by color. The distance between the circles (keywords) provides crucial information about how they are related; the weaker the relationship, the further apart the circles are. This connection is established by the frequency with the terms co-occur in the titles and abstracts [39]. According to the subject area, the VOSviewer identified four distinct clusters that could be differentiated by four distinct colors:Red cluster—“PST for stress, mental toughness, and coping”: This cluster is the largest and is composed of sixteen items as follows; stress, mental toughness, self-regulation, coping, and achievement. This cluster is associated with interventions regarding stress and coping skills and includes a perception of mental toughness for performance enhancement and questionnaire development;Green cluster—“PST for anxiety, motivation, self-confidence, and self-efficacy”: This cluster is the second largest and consists of fourteen items as follows; anxiety, athletic performance, motivation, attitude, self-confidence, and strategies. This cluster relates to a mental training program for anxiety and self-confidence of athletes in a variety of sports fields and a motivational climate for athletic performance;Blue cluster—“PST for flow and mindfulness”: the third cluster includes 11 keywords as follows; flow, imagery, intervention, mindfulness, and performance enhancement. This cluster refers to the implementation of mindfulness intervention for performance enhancement and the study of the relationship between flow, mindfulness, and PST;Yellow cluster—“PST for emotions”: the last cluster is composed of four items as follows; competition, emotions, model, and sports. This cluster mainly relates to emotional regulation and emotional intelligence in sports and athletic performance.

#### 3.4.1. Cluster 1—Red: PST for Stress, Mental Toughness, and Coping

Primarily, this cluster accumulates publications related to the major role of PST, stress management, or developing mental toughness for peak performance. Nevertheless, we found three key approaches in this cluster. First, categorized studies that include the process of identifying the source of stress and measuring it through the development of a questionnaire [40,41,42]. The second academic approach identified in this cluster is coping strategies and responses to the various sports settings and populations (e.g., soccer, volleyball, CrossFit, football, etc.) [43,44,45,46]. The last approach of the analyzed studies related to the development and maintenance of mental toughness for performance enhancement [47,48] and examining the relationship between mental toughness and other psychological skill in a variety of elite athletes [49,50].

Sources of stress in the sports setting are diverse. Specifically, Kroll [51] categorized five psychological stress that was encountered by adult athletes; fear of failure, feelings of inadequacy, loss of internal control, guilt, and current physical state. Whilst sources of stress are considerably different across individuals, related studies of a variety of populations have confirmed the generalizability of Kroll’s idea. Studies focused on soccer players [52] and basketball teams [42] consistently reported sources of stress include fear of failure, concern about the expectations of others, making mistakes, the media, and unforeseen events. The most apparent application of these results seems to be that in order for performers to have any chance of putting on a relatively stress-free performance, they must have complete confidence in their goals, their organization’s system, and their event preparation.

Although stress can be viewed as both demanding and anxiety-inducing, the approach of PST literature on stress and performance has mostly concentrated on the use of coping strategies in various events. A study investigating the strategies for table tennis players coping with their anxiety level reported that the coping strategies include PST techniques such as positive self-talk, breathing techniques, and visualization to enhance their performance [53]. In the same year, Vidic and colleagues [42] published a mixed methodology study investigating the PST intervention effect on women’s NCAA division basketball player’s perceived stress and athletic coping skills. Both quantitative and qualitative results showed a progressive decrease in stress and an increase in athletic coping skills across the study. Moreover, their study opened up new possibilities for future research by incorporating mindfulness into the PST program.

Since the early 2000s, this area of knowledge has attracted the interest of sports psychologists, coaches, and professionals, as mental toughness is an essential aspect of success in sports. In the early stage of the mental toughness research, based on the studies that identify the mental toughness of athletes in various sports events [48,54], the development and validation of inventory to measure and maintain mental toughness continued [55,56]. Currently, mental toughness is a frequently studied topic, along with other topics such as anxiety, psychological skills, and mindfulness. For instance, a prior study in our search collection examined the psychological skills, mental toughness, and degree of anxiety of female football players in relation to their skill level [57]. The psychological skills of the athletes did not differ according to skill level, but in terms of mental toughness and anxiety, the national team scored highest and lowest, respectively. Similarly to this, Wu and colleagues [58] explored the connection between psychological skills, dispositional mindfulness, and mental toughness among collegiate athletes. The authors discovered a beneficial relationship between mindfulness, mental toughness, and psychological skills in relation to sports performance, and they offered potential directions for future studies to improve both athletic performance and quality of life.

#### 3.4.2. Cluster 2—Green: PST for Anxiety, Motivation, Self-Confidence, and Self-Efficacy

Cluster 2, represented by the green color, includes 14 keywords. This cluster consisted of publications related to the PST interventions that athlete’s anxiety and motivation climate in various sports events. The papers that fit into our cluster can be classified into two distinct groups: (i) influence or relationship between anxiety and self-confidence in athletic performance and (ii) motivational climate and self-efficacy in athletes.

Goal setting, self-talk, pre-shot routine, relaxation, and imagery are the most popular mental techniques employed by professional athletes to improve their performance [59]. In terms of the techniques employed by the coaches to promote their players’ self-efficacy beliefs, according to Weinberg and Jackson [60], they were promoting positive self-talk, acting as a role model for confidence, and verbal praise and persuading. Ultimately, these techniques are based on the basic hypothesis that PST modulates anxiety and confidence to a certain degree in athletes. However, regarding the extent to which these programs can be successful in assisting athletes in controlling their anxiety and confidence, there was a lack of empirical evidence. Since then, through the two decades, numerous study has presented scientific evidence on the effect of PST on the control of self-confidence and anxiety. As a representative study, Terry, Coakley, and Karageorghis [61] studied the relevance of matching hypotheses for anxiety interventions in junior tennis players. The finding rejected the matching hypothesis and demonstrated that, while all techniques were effective in decreasing cognitive anxiety and increasing self-confidence, centering was the most effective treatment for lowering cognitive anxiety, while mental rehearsal was more successful in decreasing somatic anxiety. The only trait for which the combined intervention outperformed either centering or mental rehearsal alone was self-confidence. Furthermore, Thomas et al. [62] asserted that anxiety symptoms vary depending on personal interpretation. They measured the associated anxiety symptoms of facilitators (i.e., a performer with a positive interpretation of both cognitive and somatic symptoms) and debilitators across the dimensions of intensity, direction, and frequency throughout the 7-day competition cycle. The results showed that facilitators, interpreting their anxiety symptoms as positive toward performance, experienced higher self-confidence throughout the pre-competition period.

In general, the term “motivational climate” refers to the psychological environment that coaches, parents, and teammates primarily create for their athletes when they train and compete. Most studies on the motivational climate in sports settings have emphasized the importance of the coach in creating a positive sports environment [63]. Young athletes may improve their skill level, increase performance efficiency in competition, and develop not just as athletes but as people in a positive motivational environment [64]. According to athletes’ perceptions of the accomplishment environment, which are based on the Achievement Goal Theory (AGT), a related study confirmed two main motivational climates. The task- and ego-involving environments described by AGT can coexist or be combined [65]. Ames claims that the term “motivational climate” refers to how athletes understand the context-specific cues, rules, and expectations that enable the dissemination of task- and ego-involving motivational cues that support the formation of certain goal orientations.

On the other hand, the impact of various motivational climates on emotional, cognitive, and motivational processes has received a lot of scientific attention. Improvements in enjoyment, effort, perceived competence, and self-efficacy were connected to adaptive cognitive and emotional processes in a mastery setting. A performance climate, on the other hand, was linked to less adaptive behavior, including increased performance anxiety and concern [66]. In one of the earliest investigations into the relationship between motivational climate and self-efficacy, Wood and Bandura [67] found that a task-involving climate was linked to higher levels of self-efficacy and performance when people encountered difficulty as opposed to an ego-involving climate.

#### 3.4.3. Cluster 3—Blue: PST for Flow and Mindfulness

In cluster 3, represented by the color green, there are eleven keywords, such as flow, mindfulness, and performance enhancement. This cluster included papers that examine specific links between psychological skills and strategies, mental state of flow, and optimal performance, as well as the relationship between flow and mindfulness. The investigation of these relationships serves the aims of expanding the research of antecedent of flow state in sports, as well as examining the relationship between mindfulness and quality of athletic performance.

Research on flow in sports increased in the early 1990s [68,69], and Csikszentmihalyi [70] encouraged the application of flow theory to sports settings, where some of his early flow research began. Theoretically, flow, as an optimal mental state, is expected to be related to optimal athletic performance, as well as delivering an optimal experience. A preliminary study that examines the conditions or factors that are positively associated with athletes being able to attain flow failed to identify the relationship between psychological constructs and attainment of flow in recreational sports settings [71]. However, more recently, Jackson et al. found positive and negative associations between flow and intrinsic motivation, perceived ability, and cognitive anxiety, respectively [72]. Flow is typically viewed as a peak performance state, and there is some evidence to support this assumption [68]. Nonetheless, further study is required to investigate the link between flow and performance in sports.

There has been little research investigating the relationship between mindfulness and the adoption of mental skills in sports. In the initiation of mindfulness study among athletes, Gardner and Moore [73] reported two case studies illustrating the potential efficacy of their mindfulness-based intervention program, which they named the Mindfulness-Acceptance-Commitment approach. The author claim that planned self-regulation of present-moment awareness training that includes mindfulness awareness of breath and bodily movements enhanced participants’ athletic performance and enjoyment. In particular, the acceptance of negative thoughts reduced worrying, improved enjoyment, concentration, and persistence are some of the beneficial results of Gardner and Moore’s mindfulness-based intervention program. Starting with Gardner and Moore’s study, theoretical and methodological considerations regarding mindfulness have been made, and the scope of research is expanding to strength training [74], the sports field [42], and brain science [75]. More recently, efforts to increase mental toughness and psychological well-being through mindfulness programs have continued [76], and mindfulness programs are being used as a way to protect athletes from psychological distress, especially during COVID-19 [77]. Unfortunately, there is also a lack of research examining the relationship between flow and mindfulness. Although Clark [78] examined the impact of mindfulness training on the time spent in flow based on a non-athlete sample, there is only one study conducted on an athlete sample in this search collection.

#### 3.4.4. Cluster 4—Yellow: PST for Emotions

This area of study is mainly related to emotional regulation or emotional intelligence during sports performance and emotional regulation strategies. Research on emotion in the field of sports was initiated in the late 2000s, and vigorous research has been conducted since 2010.

Emotional Intelligence (EI) indicates individual responses to intrapersonal or interpersonal emotional information and includes the recognition, expression, comprehension, and modulation of personal and other’s emotions [79]. There is a growing body of research that suggests that EI has a crucial role in athletic performance and physical activity [80]. To enhance sports performance or exercise adherence, a critical understanding of EI constructs is especially important to practicing consultants targeting the implementation of evidence-based intervention. Specifically, two studies that had investigated EI as it relates to psychological skill usage [81,82] in this search collection. A higher trait EI was shown to be related to more frequent use of psychological skills. In one study, high trait EI was associated with more use of task-oriented coping methods, and task-oriented coping is frequently characterized as the most effective coping style for successful sports performance [80]. Furthermore, stronger athletic success motivation (the capacity to efficiently push oneself toward sports performance) has been linked to higher trait EI [82]. A more direct examination of psychological skill utilization among athletes discovered that higher scores on trait EI components were associated with more frequent use of self-talk, imagery, emotional regulation, goal setting, activation, and relaxation strategies in practice and competition [81]. A recent study examined the relationship between EI and anxiety, motivation, and leadership in athletes using a structural equation model. The study shows a strong and direct relationship between EI and anxiety and between EI and motivation, but the direct relationship between EI and leadership is not identified [83].

The automatic or purposeful use of strategies to elicit, preserve, change, or express emotions is known as emotion regulation [84]. If an athlete believes that controlling their emotions will help them perform better, they are more likely to attempt. Athletes build meta-beliefs about the emotions necessary for peak performance, and these beliefs are important for emotion control during competition [85]. Specifically, many athletes like feeling nervous before a competition and will up-regulate that emotion accordingly [85]. A recent study that investigated emotion regulation strategies used in endurance athletes reported that meta-emotion beliefs that methods targeted at raising anxiety and/or anger would benefit performance, and they utilized techniques to raise the strength of those emotions; that is, they tried to make themselves feel angrier or anxious in order to improve performance [86]. Furthermore, athletes use many thoughts and behaviors to regulate their emotions. Robazza et al. [87] found that athletes utilized various emotion regulation strategies, such as self-talk and imagery. An advantage of using the Individual Zone of Optimal Functioning (IZOF) approach in sports (i.e., unpleasant but helpful for achieving success in competition) is that it facilitates detailed analysis of specific instance of real-world experience that has practical value as well as facilitate how theory can be observed in an ecologically plausible environment. In the research field of emotion regulation in sports, various theoretical approaches are being attempted to gain the broadest possible understating of the psychological and behavioral aspects of athletes.

## 4. Conclusions, Limitations, and Future Research

Our understanding of the existing situation and evolution of PST in sports is assisted by the study’s findings. This information is significant because it gives a comprehensive view of the publications, authors, institutions, and journals with the greatest number of publications and citations, as determined by an examination of 405 total articles. In this study, thematic areas where PST-related studies are flourishing in sports were identified through bibliographic analysis. Basically, traditional topics such as stress management, anxiety control, and coping were the main psychological techniques for optimal performance. In addition, confidence, self-efficacy, and mental toughness were identified as major topics for psychological factors for optimal performance. In addition, to improve athletes’ quality of life, PST programs include mindfulness, emotions, and positive psychology. This study allows not only to determine the topics and areas of interest for authors and academics but also to figure out future research associated with the development and state of each cluster.

It is important to acknowledge any potential limitations of this study. Although the Web of Science database was utilized for the search, which was commonly used in earlier research [22,88] and is extensively used for academic searches, not all of the pertinent material may have been covered in our investigation. Similarly, a qualitative analysis was conducted to decide which papers to include or omit in this study. Although this process may have taken into account the authors’ biases, it enhanced the credibility of the study results by removing studies that mentioned PST for firefighters., surgeons, and military soldiers and including only those referring to PST in sports/elite athletes. For future research, a comparison of these results is recommended with those from other databases such as Scopus, EBSCO, or Google Scholar. In addition, it is suggested to conduct a qualitative analysis of the search results so that useful information can be obtained for academics and experts. These types of studies can provide detailed information about gaps in the existing literature. Sport, especially in the field of PST, is going through a phase of change from a consumer-centric perspective, so it is important to focus attention on theoretical and empirical developments.

## Figures and Tables

**Figure 1 healthcare-11-00259-f001:**
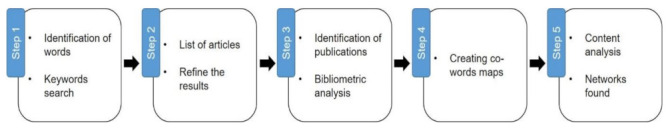
Schematic process of the bibliometric analysis.

**Figure 2 healthcare-11-00259-f002:**
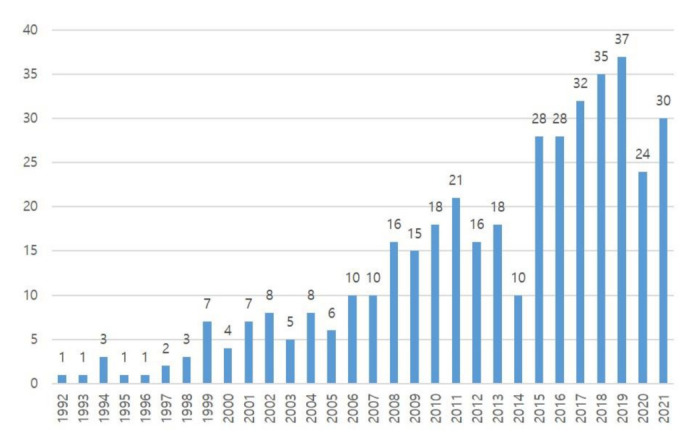
Number of articles published per year (1992–2021).

**Figure 3 healthcare-11-00259-f003:**
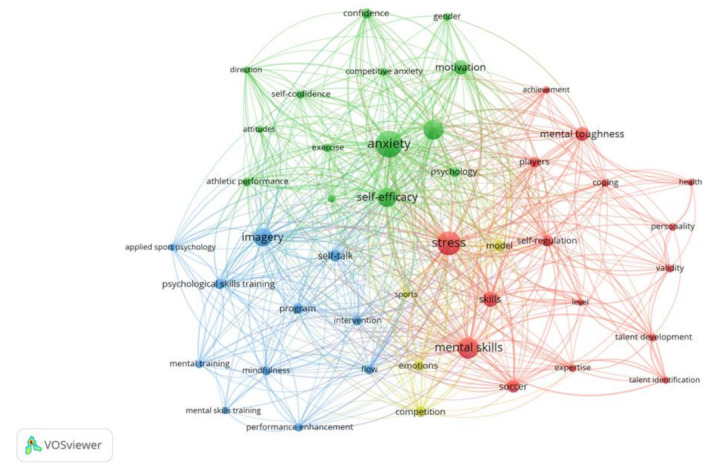
Co-word network created by VOSviewer.

**Table 1 healthcare-11-00259-t001:** Authors with the highest number of publications (>5).

Author	Affiliation	No.	LCS	GCS
Lane, AM	Wolverhampton University (UK)	13	23	411
Collins, D	University Central Lancashire (UK)	10	36	295
Devonport, TJ	Wolverhampton University (UK)	9	25	294
Hanton, S	Cardiff Metropolitan University (UK)	8	75	439
MacNamara, A	University Central Lancashire (UK)	7	34	241
Grobbelaar, HW	Stellenbosch University (South Africa)	6	9	35
Hardy, L	Bangor University (UK)	6	103	367
Maynard, I	Sheffield Hallam University (UK)	6	21	228
Ortega, E	Singapore University of Social Sciences (Singapore)	6	3	71

No.: number of articles, LCS: local citation score, GCL: global citation score.

**Table 3 healthcare-11-00259-t003:** Number of publications by Institutions.

Institution	Country	No.	LCS	GCS
Wolverhampton University	UK	13	37	432
University Central Lancashire	UK	11	35	288
Sheffield Hallam University	UK	9	23	259
Bangor University	UK	8	57	266
Cardiff Metropolitan University	UK	8	6	60
Staffordshire University	UK	8	17	223
University of Portsmouth	UK	8	42	222
The University of Queensland	Australia	8	10	74
University Of Wales Institute Cardiff	UK	8	79	472

No.: number of articles, LCS: local citation score, GCL: global citation score.

**Table 4 healthcare-11-00259-t004:** Journals by the Number of Publications and Citation Received (LCS and GCS) and Impact Factor (JCR).

Journal	No.	LCS	GCS	JCR (2021)
Sport Psychologist	58	183	1465	1.46 (Q4)
Journal of Applied Sport Psychology	32	181	1177	3.36 (Q2)
Frontiers in Psychology	26	0	253	4.23 (Q1)
Psychology of Sport and Exercise	26	54	735	5.11 (Q1)
Journal of Sports Sciences	14	132	876	3.94 (Q2)
International Journal of Sport Psychology	11	5	74	0.66 (Q4)
Journal of Sports Science and Medicine	11	35	322	3.84 (Q2)

No.: number of articles, LCS: local citation score, GCL: global citation score.

**Table 5 healthcare-11-00259-t005:** Most frequent keywords.

Most Frequent Keywords (≥50)				Most Cited Keywords (≥1000)			
Keyword	*f*	LCS	GCS	Keyword	*f*	LCS	GCS
Psychological	129	357	2298	Performance	103	425	3333
Skill	116	393	2121	Psychological	129	375	2298
Performance	103	425	3333	Sport	86	144	2157
Athlete	95	219	1650	Skill	116	393	2121
Sport	86	144	2157	Mental	75	259	1849
Mental	75	259	1849	Athlete	95	219	1650
Training	72	213	973	Review	21	75	1320
Elite	50	96	1015	Elite	50	96	1015

*f*: frequency; LCS: local citation score; GCL: global citation score.

## Data Availability

The data presented in this study are available on request from the corresponding author. The data are not publicly available due to WoS policy.

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
