# Peer review of "Psychological Skills Training for Athletes in Sports: Web of Science Bibliometric Analysis"

_healthcare, 2023, doi:10.3390/healthcare11020259_

Round 1
Reviewer 1 Report
Dear authors,
The topic “Psychological Skills Training for Athlete in Sports: Web of Science Bibliometric Analysis” is an interesting and essential study. The manuscript provided meaningful findings for athletes and coaches. Here are several comments for the manuscript.
1. The abstract is not well-written. Be sure to write the abstract more briefly. Also, I suggest adding some information about the results.
2. I suggest add Sport into keywords.
3. Technical depth of the paper is limited and should be improved.
4. Result section is weak and can be improved with more comparative
analysis in the manuscript.
5. The manuscript should be strengthened with recent Reference.
6. Does the coding process has co-coders? If yes, please describe the inter coders reliability.
7. The study needs to mention the reliability and validity of the qualitative analysis process.
8. I suggest the results should be written in the present tense. Some sentences are past tense, and some are present tense. Please be consistent.
9. I also suggest the results should be added the percentage in the tables when the study used content analysis.
10.Need to justify for selecting the abstracts between 1992 and 2021? Can it be a random period? Any time-series analysis?
11.Usually, UCINET as network analysis software. Why this study uses VOSviewer in network analysis.
12.In Figure 3, they describe in general terms what they did, but not how they did it. It is important to specify the software they used.
Author Response
We appreciate the careful and detailed review of the manuscript with several helpful suggestions that we have incorporated into the revised manuscript. The response to each comment from the reviewer is described below along with a summary of the changes made, which are indicated with tracks in the revised manuscript.
Comments from the Editors and Reviewers:
Reviewer #1:
Dear authors,
The topic “Psychological Skills Training for Athlete in Sports: Web of Science Bibliometric Analysis” is an interesting and essential study. The manuscript provided meaningful findings for athletes and coaches. Here are several comments for the manuscript.
- The abstract is not well-written. Be sure to write the abstract more briefly. Also, I suggest adding some information about the results.
We appreciate the reviewer’s comments. As the reviewer requested, we revised the abstract.
- I suggest add Sport into keywords.
Added.
- Technical depth of the paper is limited and should be improved.
We agree with the reviewer’s idea. We added a technical statements of visualized map. Please see the lines 117-130.
- Result section is weak and can be improved with more comparative analysis in the manuscript.
As the reviewer commented, we tried to strengthen the result section by adding more comparative and recent studies.
- The manuscript should be strengthened with recent Reference.
This study is intended to organize the process from the early stage of PST research to the latest study, and contains a large amount of references. Nevertheless, up-to-date references were added at the request of reviewers.
- Does the coding process has co-coders? If yes, please describe the inter coders reliability.
Bibliometrics are the quantitative analysis of academic publications. Using academic publications as a data source, bibliometric analysis attempts to provide a better understanding of how research is produced, organized, and interrelated. Thus, this study is nothing to do with coding process.
- The study needs to mention the reliability and validity of the qualitative analysis process.
As we stated in #6, reliability and validity are nothing to do with this study.
- I suggest the results should be written in the present tense. Some sentences are past tense, and some are present tense. Please be consistent.
We thoroughly read through the results section. In the results, we used the past tense only when referring to the results of the previous studies or when expressing the passive voice.
- I also suggest the results should be added the percentage in the tables when the study used content analysis.
As we noted in #6, this study is not a content analysis.
10.Need to justify for selecting the abstracts between 1992 and 2021? Can it be a random period? Any time-series analysis?
In order to obtain as much data as possible, we did not put a specific time period limit on the search. The WOS search engine automatically defined the time period.
11.Usually, UCINET as network analysis software. Why this study uses VOSviewer in network analysis.
Among VOSviewer, UCINET, Netminer, and Pajek, VOSviewer and Pajek are most widely used programs because they are freeware, so they are highly useful and easy to access. VOSviewer is software specialized for visualization of bibliographic analysis and clustering analysis.
12.In Figure 3, they describe in general terms what they did, but not how they did it. It is important to specify the software they used.
I believe a sufficient explanation of how circles are made and their sizes, distances between circles, and colors are determined in the software is provided. Furthermore, an explanation of how the connection between the two circles is created was also provided. Of course, there are logic, formulas, and algorithms that generate these results before that, but I think the explanation of how these formulas or algorithms are working is beyond the scope of this study.
Reviewer 2 Report
1. Keywords: I suggest changing the keywords because using keywords different from the title can more the possibility for the article to be found. (line 24)
Keywords: psychological skills; psychological sport; psychological training; mental training.
2. Objective of the study - It is important to make the purpose of the study clear. It helps the readers understand the conclusion of the study. (line 77)
3. Conclusion
a. The term "Perhaps" is vague, not saying anything. Do not term adequate for use in the conclusion section. (line 449).
b. In these lines is a major conclusion, but it is partial. But, it seems the authors are discussing. The authors need to deepen the conclusion and to write in a way affirmative and relationship to the objective of the study. (lines 450 to 456).

Author Response
We appreciate the careful and detailed review of the manuscript with several helpful suggestions that we have incorporated into the revised manuscript. The response to each comment from the reviewer is described below along with a summary of the changes made, which are indicated with tracks in the revised manuscript.
Comments from the Editors and Reviewers:
- Keywords: I suggest changing the keywords because using keywords different from the title can more the possibility for the article to be found. (line 24)
Keywords: psychological skills; psychological sport; psychological training; mental training.
We appreciate your comment, Keywords revised as reviewer recommended.
- Objective of the study - It is important to make the purpose of the study clear. It helps the readers understand the conclusion of the study. (line 77)
We agree with the reviewer’s idea, we revised the objective of the study as reviewer’s comments.
- Conclusion
- The term "Perhaps" is vague, not saying anything. Do not term adequate for use in the conclusion section. (line 449).
We agree with the reviewer’s point of view and delete the term.
- In these lines is a major conclusion, but it is partial. But, it seems the authors are discussing. The authors need to deepen the conclusion and to write in a way affirmative and relationship to the objective of the study. (lines 450 to 456).
Reviewer is correct. We have reorganized the conclusion and re-write the statements.